# GCOMB: Learning Budget-constrained Combinatorial Algorithms over Billion-sized Graphs

**Sahil Manchanda, Akash Mittal,**\* **Anuj Dhawan**\*
Indian Institute of Technology Delhi
{sahil.manchanda,cs1150208,Anuj.Dhawan.cs115}@cse.iitd.ac.in

**Sourav Medya**[1]**, Sayan Ranu**[2]**, Ambuj Singh**[3]
[1]Northwestern University,[2]Indian Institute of Technology Delhi
[3]University of California Santa Barbara
[1]sourav.medya@kellogg.northwestern.edu
[2]sayanranu@cse.iitd.ac.in,[3]ambuj@ucsb.edu

## Abstract

There has been an increased interest in discovering heuristics for combinatorial problems on graphs through machine learning. While existing techniques have primarily focused on obtaining high-quality solutions, scalability to billion-sized graphs has not been adequately addressed. In addition, the impact of budget-constraint, which is necessary for many practical scenarios, remains to be studied. In this paper, we propose a framework called GCOMB to bridge these gaps. GCOMB trains a Graph Convolutional Network (GCN) using a novel *probabilistic greedy* mechanism to predict the quality of a node. To further facilitate the combinatorial nature of the problem, GCOMB utilizes a *Q*-learning framework, which is made efficient through *importance sampling*. We perform extensive experiments on real graphs to benchmark the efficiency and efficacy of GCOMB. Our results establish that GCOMB is 100 times faster and marginally better in quality than state-of-the-art algorithms for learning combinatorial algorithms. Additionally, a case-study on the practical combinatorial problem of Influence Maximization (IM) shows GCOMB is 150 times faster than the specialized IM algorithm IMM with similar quality.

## 1   Introduction and Related Work

Combinatorial optimization problems on graphs appear routinely in various applications such as viral marketing in social networks [14, 4], computational sustainability [8], health-care [33], and infrastructure deployment [20, 23, 24, 22]. In these *set combinatorial problems*, the goal is to identify the set of nodes that optimizes a given objective function. These optimization problems are often NP-hard. Therefore, designing an exact algorithm is infeasible and polynomial-time algorithms, with or without approximation guarantees, are often desired and used in practice [13, 31]. Furthermore, these graphs are often dynamic in nature and the approximation algorithms need to be run repeatedly at regular intervals. Since real-world graphs may contain millions of nodes and edges, this entire process becomes tedious and time-consuming.

To provide a concrete example, consider the problem of viral marketing on social networks through *Influence Maximization* [2, 14]. Given a budget $b$, the goal is to select $b$ nodes (users) such that their endorsement of a certain product (ex: through a tweet) is expected to initiate a cascade that reaches the largest number of nodes in the graph. This problem is NP-hard [14]. Advertising through social networks is a common practice today and needs to solved repeatedly due to the graphs being dynamic

in nature. Furthermore, even the greedy approximation algorithm does not scale to large graphs [2] resulting in a large body of research work [31, 13, 16, 26, 14, 5, 32, 6].

At this juncture, we highlight two key observations. First, although the graph is changing, the underlying model generating the graph is likely to remain the same. Second, the nodes that get selected in the answer set of the approximation algorithm may have certain properties common in them. Motivated by these observations, we ask the following question [7]: *Given a set combinatorial problem P on graph G and its corresponding solution set S, can we learn an approximation algorithm for problem P and solve it on an unseen graph that is similar to G?*

## 1.1 Limitations of Existing Work

The above observations were first highlighted by S2V-DQN [7], where they show that it is indeed possible to *learn* combinatorial algorithms on graphs. Subsequently, an improved approach was proposed in GCN-TREESEARCH [19]. Despite these efforts, there is scope for further improvement.

• **Scalability**: The primary focus of both GCN-TREESEARCH and S2V-DQN have been on obtaining quality that is as close to the optimal as possible. Efficiency studies, however, are limited to graphs containing only hundreds of thousands nodes. To provide a concrete case study, we apply GCN-TREESEARCH for the Influence Maximization problem on the YouTube social network. We observe that GCN-TREESEARCH takes one hour on a graph containing a million edges (Fig. 3a; we will revisit this experiment in § 4.3). Real-life graphs may contain billions of edges (See. Table 1a).

• **Generalizability to real-life combinatorial problems:** GCN-TREESEARCH proposes a learning-based heuristic for the Maximal Independent Set problem (MIS). When the combinatorial problem is not MIS, GCN-TREESEARCH suggests that we map that problem to MIS. Consequently, for problems that are not easily mappable to MIS, the efficacy is compromised (ex: Influence Maximization).

• **Budget constraints:** Both GCN-TREESEARCH and S2V-DQN solve the decision versions of combinatorial problems (Ex. set cover, vertex cover). In real life, we often encounter their budget-constrained versions, such as max-cover and Influence Maximization [14].

Among other related work, Gasse et al. [9] used GCN for learning branch-and-bound variable selection policies, whereas Prates et al. [27] focused on solving Travelling Salesman Problem. However, the proposed techniques in these papers do not directly apply to our setting of set combinatorial problems.

## 1.2 Contributions

At the core of our study lies the observation that although the graph may be large, only a small percentage of the nodes are likely to contribute to the solution set. Thus, pruning the search space is as important as prediction of the solution set. Both S2V-DQN [7] and GCN-TREESEARCH [19] have primarily focused on the prediction component. In particular, S2V-DQN learns an end-to-end neural model on the entire graph through reinforcement learning. The neural model integrates node embedding and $Q$-learning into a single integrated framework. Consequently, the model is bogged down by a large number of parameters, which needs to be learned on the entire node set. As a result, we will show in §. 4 that S2V-DQN fails to scale to graphs beyond $20,000$ nodes.

On the other hand, GCN-TREESEARCH employs a two-component framework: **(1)** a graph convolutional network (GCN) to learn and predict the individual *value* of each node, and **(2)** a *tree-search* component to analyze the dependence among nodes and identify the solution set that collectively works well. Following tree-search, GCN is repeated on a reduced graph and this process continues iteratively. This approach is not scalable to large graphs since due to repeated iterations of GCN and TreeSearch where each iteration of tree-search has $O(|E|)$ complexity; $E$ is the set of edges.

Our method GCOMB builds on the observation that computationally expensive predictions should be attempted only for promising nodes. Towards that end, GCOMB has two separate components: **(1)** a GCN to prune *poor* nodes and learn embeddings of *good* nodes in a *supervised* manner, and **(2)** a $Q$-learning component that focuses only on the *good* nodes to predict the solution set. Thus, unlike S2V-DQN, GCOMB uses a *mixture* of supervised and reinforcement learning, and does not employ an end-to-end architecture. Consequently, the prediction framework is lightweight with a significantly reduced number of parameters.

When compared to GCN-TREESEARCH, although both techniques use a GCN, in GCOMB, we train using a novel *probabilistic greedy* mechanism. Furthermore, instead of an iterative procedure of repeated GCN and TreeSearch calls, GCOMB performs a single forward pass through GCN

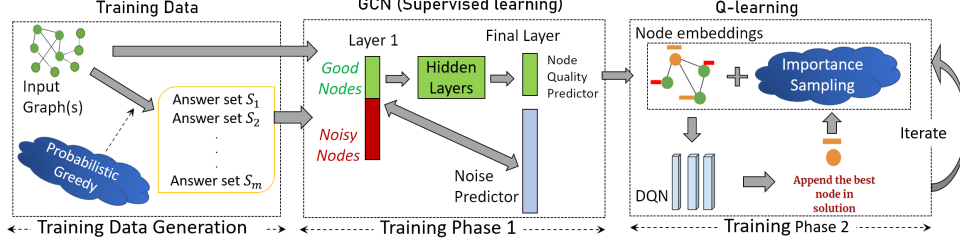

Figure 1: The flowchart of the training phase of GCOMB.

during inference. In addition, unlike TreeSearch, which is specifically tailored for the MIS problem, GCOMB is problem-agnostic [2]. Finally, unlike both S2V-DQN and GCN-TREESEARCH, GCOMB uses lightweight operations to prune *poor* nodes and focus expensive computations only on nodes with a high potential of being part of the solution set. The pruning of the search space not only enhances scalability but also removes noise from the search space leading to improved prediction quality. Owing to these design choices, **(1)** GCOMB is scalable to billion-sized graphs and up to $100$ times faster, **(2)** on average, computes higher quality solution sets than S2V-DQN and GCN-TREESEARCH, and **(3)** improves upon the state-of-the-art algorithm for Influence Maximization on social networks.

## 2   Problem Formulation

**Objective:** *Given a budget-constrained set combinatorial problem $P$ over graphs drawn from distribution $D$,* learn *a heuristic to solve problem $P$ on an unseen graph $G$ generated from $D$.*

Next, we describe three instances of budget-constrained set combinatorial problems on graphs.

**Maximum Coverage Problem on bipartite graph (MCP):** Given a bipartite graph $G = (V, E)$, where $V = A \cup B$, and a budget $b$, find a set $S^* \subseteq A$ of $b$ nodes such that coverage is maximized. The coverage of set $S^*$ is defined as $f(S^*) = \frac{|X|}{|B|}$, where $X = \{j|(i,j) \in E, i \in S^*, j \in B\}$.

**Budget-constrained Maximum Vertex Cover (MVC):** Given a graph $G = (V, E)$ and a budget $b$, find a set $S^*$ of $b$ nodes such that the coverage $f(S^*)$ of $S^*$ is maximized. $f(S^*) = \frac{|X|}{|E|}$, where $X = \{(i,j)|(i,j) \in E, i \in S^*, j \in V\}$.

**Influence Maximization (IM) [2]:** Given a budget $b$, a social network $G$, and a information diffusion model $\mathcal{M}$, select a set $S^*$ of $b$ nodes such that the expected diffusion spread $f(S^*) = \mathbb{E}[\Gamma(S^*)]$ is maximized. (See App. A in supplementary for more details).

## 3   GCOMB

The input to the training phase is a set of graphs and the optimization function $f(\cdot)$ corresponding to the combinatorial problem in hand. The output is a sequence of two separate neural graphs, GCN [10] and $Q$-learning network, with their corresponding learned parameters $\Theta_G$ and $\Theta_Q$ respectively. In the testing phase, the inputs include a graph $G = (V, E)$, the optimization function $f(\cdot)$ and the budget $b$. The output of the testing part is the solution set of nodes constructed using the learned neural networks. Fig. 1 presents the training pipeline. We will now discuss each of the phases.

### 3.1   Generating Training Data for GCN

Our goal is to learn node embeddings that can predict "quality", and thereby, identify those nodes that are likely to be part of the answer set. We could adopt a classification-based method, where, given a training graph $G = (V, E)$, budget $b$ and its solution set $S$, a node $v$ is called *positive* if $v \in S$; otherwise it is negative. This approach, however, assumes all nodes that are not a part of $S$ to be equally bad. In reality, this may not be the case. Consider the case where $f(\{v_1\})=f(\{v_2\})$, but the marginal gain of node $v_2$ given $S = \{v_1\}$, i.e., $f(\{v_1, v_2\}) - f(\{v_1\})$, is 0 and vice versa. In this scenario, only one of $v_1$ and $v_2$ would be selected in the answer set although both are of equal quality on their own.

**Probabilistic greedy:** To address the above issue, we *sample* from the *solution space* in a greedy manner and learn embeddings that reflect the *marginal gain* $f(S \cup \{v\}) - f(S)$ provided by a node $v$ towards the solution set $S$ (Alg. 2 in Appendix). To sample from the solution space, in each iteration, instead of selecting the node with the highest marginal gain, we choose a node with probability proportional to its marginal gain. The probabilistic greedy algorithm runs $m$ times to construct $m$ different solution sets $\mathbb{S} = \{S_1, \cdots, S_m\}$ and the score of node $v \in V$ is set to:

$$score(v) = \frac{\sum_i^m gain_i(v)}{\sum_i^m f(S_i)} \tag{1}$$

Here, $gain_i(v)$ denotes the marginal gain contribution of $v$ to $S_i$. Specifically, assume $v$ is added to $S_i$ in the $(j+1)_{th}$ iteration and let $S_i^j$ be the set of nodes that were added in the first $j$ iterations while constructing $S_i$. Then, $gain_i(v) = f\left(S_i^j \cup \{v\}\right) - f\left(S_i^j\right)$. In our experiments, $m$ is set to 30 for all three problems of MCP, MVC and IM.

**Termination condition of probabilistic greedy:** Probabilistic greedy runs till *convergence* of the marginal gains, i.e., $gain_i(v) \leq \Delta$, where $\Delta$ is a small value. The goal here is to identify all nodes that could potentially be part of the solution set for *any* given budget. $\Delta$ in our experiments is set to 0.01 for all three problems of MCP, MVC and IM.

## 3.2 Training the GCN

Our goal in this phase is two-fold: **(1)** Identify nodes that are unlikely to be part of the solution set and are therefore noise in the context of our problem; **(2)** Learn a predictive model for node quality.

**Noise predictor:** The noise predictor should be lightweight so that expensive computations are reserved only for the good nodes. With this goal, we exploit the first layer information of the GCN and learn a classifier to predict for a given budget $b$, whether a node can be safely pruned without affecting the quality of the solution set. Typically, the first layer of a GCN contains the raw features of nodes that are relevant for the problem being solved. In GCOMB, we use the summation of the outgoing edge weights as node features. Let $\boldsymbol{x}_v$ denote the total outgoing edge weight of node $v$. To learn the noise predictor, given a set of training graphs $\{G_1, \cdots, G_t\}$, we first sort all nodes based on $\boldsymbol{x}_v$. Let $rank(v, G_i)$ denote the position of $v$ in the sorted sequence based on $\boldsymbol{x}_v$ in $G_i$. Furthermore, let $S_j^i$ denote the $j^{th}$ solution set constructed by probabilistic greedy on $G_i$. Given a budget $b$, $S_{G_i,b}^j \subseteq S_j^i$ denotes the subset containing the first $b$ nodes added to $S_j^i$ by probabilistic greedy. Therefore, $r_{G_i}^b = \max_{j=0}^m \left\{ \max_{\forall v \in S_{G_i,b}^j} \{rank(v, G_i)\} \right\}$ represents the lowest rank of any node in a solution set of budget $b$ in $G_i$. This measure is further generalized to all training graphs in the form of $r_{max}^b = \max_{\forall G_i} \left\{ r_{G_i}^b \right\}$, which represents the lowest rank of any node that has a realistic chance of being included in an answer set of budget $b$. To generalize across budgets, we compute $r_{max}^{b_i}$ for a series of budgets $\{b_1, \cdots, b_{max}\}$, where $b_{max} = \max_{\forall G_i} \left\{ \max_{j=0}^m \{|S_j^i|\} \right\}$. On this data, we can perform curve fitting [1] to predict $r_{max}^b$ for any (unseen) budget $b$. In our experiments, we use linear interpolation. To generalize across graph sizes, all of the above computations are performed on *normalized* budgets, where $b$ is expressed in terms of the proportion of nodes with respect to the node set size of the graph. Similarly, rank $rank(v, G_i)$ is expressed in terms of percentile.

**Node quality predictor:** To train the GCN, we sample a training graph $G_i = (V_i, E_i)$ and a (normalized) budget $b$ from the range $(0, b_{max}^i]$, where $b_{max}^i = \max_{j=0}^m \left\{ \frac{|S_j^i|}{|V_i|} \right\}$. This tuple is sent to the noise predictor to obtain the good (non-noisy) nodes. The GCN parameters ($\Theta_G$) are next learned by minimizing the loss function only on the good nodes. Specifically, for each good node $v$, we want to learn embeddings that can predict $score(v)$ through a surrogate function $score'(v)$. Towards that end, we draw multiple samples of training graphs and budgets, and the parameters are learned by minimizing the *mean squared error* loss (See Alg.3 for detailed pseudocode in the Supplementary).

$$J(\Theta_G) = \sum_{\sim \langle G_i, b \rangle} \frac{1}{|V_i^g|} \sum_{\forall v \in V_i^g} (score(v) - score'(v))^2 \tag{2}$$

In the above equation, $V_i^g$ denotes the set of good nodes for budget $b$ in graph $G_i$. Since GCNs are trained through message passing, in a GCN with $K$ hidden layers, the computation graph is limited to the induced subgraph formed by the $K$-hop neighbors of $V_i^g$, instead of the entire graph.

### 3.3 Learning $Q$-function

While GCN captures the individual importance of a node, $Q$-learning [29] learns the combinatorial aspect in a budget-independent manner. Given a set of nodes $S$ and a node $v \notin S$, we predict the $n$-step reward, $Q_n(S, v)$, for adding $v$ to set $S$ (action) via the surrogate function $Q'_n(S, v; \Theta_Q)$.

**Defining the framework:** We define the $Q$-learning task in terms of state space, action, reward, policy and termination with the input as a set of nodes and their predicted scores.

• **State space:** The state space characterizes the state of the system at any time step $t$ in terms of the candidate nodes being considered, i.e., $C_t = V^g \setminus S_t$, with respect to the partially computed solution set $S_t$; $V^g$ represents the set of good nodes from a training graph. In a combinatorial problem over nodes, two factors have a strong influence: **(1)** the individual quality of a node, and **(2)** its *locality*. The quality of a node $v$ is captured through $score'(v)$. Locality is an important factor since two high-quality nodes from the same neighborhood may not be good collectively. The locality of a node $v \in C_t$ ($C_t = V^g \setminus S_t$) is defined as:

$$loc(v, S_t) = |N(v) \setminus \cup_{\forall u \in S_t} N(u)| \tag{3}$$

where $N(v) = \{v' \in V \mid (v, v') \in E\}$ are the neighbors of $v$. Note that $N(v)$ may contain noisy nodes since they contribute to the locality of $v \in V^g$. However, locality (and $q$-learning in general) is computed only on good nodes. The initial representation $\boldsymbol{\mu_v}$ of each node $v \in C_t$ is therefore the 2-dimensional vector $[score'(v), loc(v, S_t)]$. The representation of the set of nodes $C_t$ is defined as $\boldsymbol{\mu_{C_t}} = \text{MAXPOOL} \{\mu_v \mid v \in C_t\}$. $\boldsymbol{\mu_{S_t}}$ is defined analogously as well. We use MAXPOOL since it captures the best available candidate node better than alternatives such as MEANPOOL. Empirically, we obtain better results as well.

• **Action and Reward:** An action corresponds to adding a node $v \in C_t$ to the solution set $S_t$. The immediate (0-step) reward of the action is its marginal gain, i.e. $r(S_t, v) = f(S_t \cup \{v\}) - f(S_t)$.

• **Policy and Termination:** The policy $\pi(v \mid S_t)$ selects the node with the highest *predicted* $n$-step reward, i.e., $\arg \max_{v \in C_t} Q'_n(S_t, v; \Theta_Q)$. We terminate after training the model for $T$ samples.

**Learning the parameter set $\Theta_Q$:** We partition $\Theta_Q$ into three weight matrices $\Theta_1$, $\Theta_2$, $\Theta_3$, and one weight vector $\Theta_4$ such that, $Q'_n(S_t, v; \Theta_Q) = \Theta_4 \cdot \boldsymbol{\mu_{C_t, S_t, v}}$, where $\boldsymbol{\mu_{C_t, S_t, v}} = \text{CONCAT} \left( \Theta_1 \cdot \boldsymbol{\mu_{C_t}}, \Theta_2 \cdot \boldsymbol{\mu_{S_t}}, \Theta_3 \cdot \boldsymbol{\mu_v} \right)$. If we want to encode the state space in a $d$-dimensional layer, the dimensions of the weight vectors are as follows: $\Theta_4 \in \mathbb{R}^{1 \times 3d}; \Theta_1, \Theta_2, \Theta_3 \in \mathbb{R}^{d \times 2}$. $Q$-learning updates parameters in a single episode via Adam optimizer[15] to minimize the squared loss.

$$J(\Theta_Q) = (y - Q'_n(S_t, v_t; \Theta_Q))^2, \text{ where } y = \gamma \cdot \max_{v \in V^g} \{Q'_n(S_{t+n}, v; \Theta_Q)\} + \sum_{i=0}^{n-1} r(S_{t+i}, v_{t+i})$$

$\gamma$ is the *discount factor* and balances the importance of immediate reward with the predicted $n$-step future reward [29]. The pseudocode with more details is provided in the Supplementary (App. C).

#### 3.3.1 Importance Sampling for Fast Locality Computation

Since degrees of nodes in real graphs may be very high, computing locality (Eq. 3) is expensive. Furthermore, locality is re-computed in each iteration. We negate this computational bottleneck through *importance sampling*. Let $N(V^g) = \{(v, u) \in E \mid v \in V^g\}$ be the neighbors of all nodes in $V^g$. Given a sample size $z$, we extract a subset $N_z(V^g) \subseteq N(V^g)$ of size $z$ and compute locality only based on the nodes in $N_z(V^g)$. Importance sampling samples elements proportional to their importance. The *importance* of a node in $N(V^g)$ is defined as $I(v) = \frac{score'(v)}{\sum_{\forall v' \in N(V^g)} score'(v')}$.

**Determining sample size:** Let $\mu_{N(V^g)}$ be the mean importance of all nodes in $N(V^g)$ and $\hat{\mu}_{N_z(V^g)}$ the mean importance of sampled nodes. The sampling is *accurate* if $\mu_{N(V^g)} \approx \hat{\mu}_{N_z(V^g)}$.

**Theorem 1** *Given an error bound $\epsilon$, if sample size $z$ is $O\left(\frac{\log |N(V^g)|}{\epsilon^2}\right)$, then* $P\left[|\hat{\mu}_{N_z(V^g)} - \mu_{N(V^g)}| < \epsilon\right] > 1 - \frac{1}{|N(V^g)|^2}$.

**Remarks:** **(1)** The sample size grows *logarithmically* with the neighborhood size, i.e., $|N(V^g)|$ and thus scalable to large graphs. **(2)** $z$ is an inversely proportional function of the error bound $\epsilon$.

### 3.4 Test Phase

Given an unseen graph $G$ and budget $b$, we **(1)** identify the noisy nodes, **(2)** embed good nodes through a single forward pass through GCN, and **(3)** use GCN output to embed them and perform Q-learning to compute the final solution set.

| Name | $|V|$ | $|E|$ |
|---|---|---|
| Brightkite (BK) | 58.2K | 214K |
| Twitter-ego (TW-ew) | 81.3K | 1.7M |
| Gowalla (GO) | 196.5K | 950.3K |
| YouTube (YT) | 1.13M | 2.99M |
| StackOverflow (Stack) | 2.69M | 5.9M |
| Orkut | 3.07M | 117.1M |
| Twitter (TW) | 41.6M | 1.5B |
| FriendSter (FS) | 65.6M | 1.8B |

(a) Datasets from SNAP repository [18].

| Dataset | BP-500 | | | Gowalla-900 | | |
|---|---|---|---|---|---|---|
| Budget | GCOMB | Greedy | Optimal | GCOMB | Greedy | Optimal |
| 2 | **0.295** | 0.295 | 0.295 | **0.75** | 0.75 | 0.75 |
| 4 | 0.495 | 0.505 | 0.51 | 0.902 | 0.904 | 0.904 |
| 6 | 0.765 | 0.77 | 0.773 | **0.941** | 0.93 | 0.941 |
| 10 | 0.843 | 0.845 | 0.845 | **0.952** | 0.952 | 0.952 |
| 15 | **0.96** | 0.953 | 0.963 | **0.963** | 0.963 | 0.963 |
| 20 | **0.998** | 0.99 | 1 | **0.974** | 0.974 | 0.974 |
| 25 | **1** | 1 | 1 | **0.985** | 0.985 | 0.985 |
| 30 | – | – | – | **0.996** | 0.996 | 0.996 |
| 35 | – | – | – | **1** | 1 | 1 |

(b) Coverage in MCP

Table 1: In (b), the specific cases where GCOMB matches or outperforms Greedy are highlighted in bold. Gowalla-900 is a small subgraph of 900 nodes extracted from Gowalla (See App. I for details).

**Complexity analysis:** The time complexity of the test phase in GCOMB is $O\left(|V| + |V^{g,K}|\left(dm_G + m_G^2\right) + |V^g|b\left(d + m_Q\right)\right)$, where $d$ is the average degree of a node, $m_G$ and $m_Q$ are the dimensions of the embeddings in GCN and $Q$-learning respectively, $K$ is the number of layers in GCN, and $V^{g,K}$ represents the set of nodes within the $K$-hop neighborhood of $V^g$. The space complexity is $O(|V| + |E| + Km_G^2 + m_Q)$. The derivations are provided in App. D.

# 4 Empirical Evaluation

In this section, we benchmark GCOMB against GCN-TREESEARCH and S2V-DQN, and establish that GCOMB produces marginally improved quality, while being orders of magnitudes faster. The source code can be found at https://github.com/idea-iitd/GCOMB .

## 4.1 Experimental Setup

All experiments are performed on a machine running Intel Xeon E5-2698v4 processor with 64 cores, having 1 Nvidia 1080 Ti GPU card with 12GB GPU memory, and 256 GB RAM with Ubuntu 16.04. All experiments are repeated 5 times and we report the average of the metric being measured.

**Datasets:** Table 1a) lists the real datasets used for our experiments.
*Random Bipartite Graphs (BP):* We also use the synthetic random bipartite graphs from S2V-DQN [7]. In this model, given the number of nodes, they are partitioned into two sets with 20% nodes in one side and the rest in other. The edge between any pair of nodes from different partitions is generated with probability 0.1. We use BP-$X$ to denote a generated bipartite graph of $X$ nodes.

**Problem Instances:** The performance of GCOMB is benchmarked on Influence Maximization (IM), Maximum Vertex Cover (MVC), and Maximum Coverage Problem (MCP) (§ 2). Since MVC can be mapped to MCP, empirical results on MVC are included in App. M.

**Baselines:** The performance of GCOMB is primarily compared with **(1)** GCN-TREESEARCH [19], which is the state-of-the-art technique to learn combinatorial algorithms. In addition, for MCP, we also compare the performance with **(2)** *Greedy* (Alg.1 in App. B), **(3)** S2V-DQN [7], **(5)** *CELF* [17] and **(6)** the *Optimal* solution set (obtained using CPLEX [12] on small datasets). Greedy and CELF guarantees a $1 - 1/e$ approximation for all three problems. We also compare with **(6)** *Stochastic Greedy(SG)* [21] in App. L. For the problem of IM, we also compare with the state-of-the-art algorithm **(7)** *IMM* [31]. Additionally, we also compare GCOMB with **(8)** *OPIM* [30]. For S2V-DQN, GCN-TREESEARCH, IMM, and OPIM we use the code shared by the authors.

**Training:** In all our experiments, for a fair comparison of GCOMB with S2V-DQN and GCN-TREESEARCH, we train all models for 12 hours and the best performing model on the validation set is used for inference. Nonetheless, we precisely measure the impact of training time in Fig. 2a. The break-up of time spent in each of the three training phases is shown in App. G in the Supplementary.

**Parameters:** The parameters used for GCOMB are outlined in App. H and their impact on performance is analyzed in App. N. For S2V-DQN and GCN-TREESEARCH, the best performing parameter values are identified using grid-search. In IMM, we set $\epsilon = 0.5$ as suggested by the authors. In OPIM, $\epsilon$ is recommended to be kept in range $[0.01, 0.1]$. Thus, we set it to $\epsilon = 0.05$.

## 4.2 Performance on Max Cover (MCP)
We evaluate the methods on both synthetic random bipartite (BP) graphs as well as real networks.
**Train-Validation-Test split:** While testing on any synthetic BP graph, we train and validate on five

| Graph | S2V-DQN | GCN-TS | GCOMB | Greedy |
|---|---|---|---|---|
| **BP-2k** | 0.87 | 0.86 | **0.89** | 0.89 |
| **BP-5k** | 0.85 | 0.84 | **0.86** | 0.86 |
| **BP-10k** | 0.84 | 0.83 | **0.85** | 0.85 |
| **BP-20k** | NA | 0.82 | **0.83** | 0.83 |

(a) Coverage achieved in MCP at $b = 15$.

| Budget | Speed-up |
|---|---|
| **20** | 4 |
| **50** | 3.9 |
| **100** | 3.01 |
| **150** | 2.11 |
| **200** | 2.01 |

(b) Speed-up against CELF in MCP on YT.

Table 2: (a) Coverage on Random Graphs in MCP. (b) Speed-up achieved by GCOMB against CELF on YT in MCP.

BP-1k graphs each. For real graphs, we train and validate on BrightKite (BK) (50 : 50 split for train and validate) and test on other real networks. Since our real graphs are not bipartite, we convert it to one by making two copies of $V$: $V_1$ and $V_2$. We add an edge from $u \in V_1$ to $u' \in V_2$ if $(u, u') \in E$.

**Comparison with Greedy and Optimal:** Table 1b presents the achieved coverage (Recall § 2 for definition of coverage). We note that Greedy provides an empirical approximation ratio of at least 99% when compared to the optimal. This indicates that in larger datasets where we are unable to compute the optimal, Greedy can be assumed to be sufficiently close to the optimal. Second, GCOMB is sometimes able to perform even better than greedy. This indicates that $Q$-learning is able to learn a more generalized policy through *delayed* rewards and avoid a myopic view of the solution space.

**Synthetic Datasets:** Table 2a presents the results. GCOMB and Greedy achieves the highest coverage consistently. While S2V-DQN performs marginally better than GCN-TREESEARCH, S2V-DQN is the least scalable among all techniques; it runs out of memory on graphs containing more than $20,000$ nodes. As discussed in details in § 1.2, the non-scalability of S2V-DQN stems from relying on an architecture with significantly larger parameter set than GCOMB or GCN-TREESEARCH. In contrast, GCOMB avoids noisy nodes, and focuses the search operation only on the good nodes.

*Impact of training time:* A complex model with more number of parameters results in slower learning. In Fig. 2a, we measure the coverage against the training time. While GCOMB's performance saturates within 10 minutes, S2V-DQN and GCN-TREESEARCH need 9 and 5 hours respectively for training to obtain its best performance.

**Real Datasets:** Figs. 2b and 2c present the achieved Coverage as the budget is varied. GCOMB achieves similar quality as Greedy, while GCN-TREESEARCH is marginally inferior. The real impact of GCOMB is highlighted in Figs. 2d and 2e, which shows that GCOMB is up to 2 orders of magnitude faster than GCN-TREESEARCH and 10 times faster than Greedy. Similar conclusion can also be drawn from the results on Gowalla dataset in App. K in Supplementary.

**Comparison with CELF:** Table 2b presents the speed-up achieved by GCOMB against CELF. The first pass of CELF involves sorting the nodes, which has complexity $O(|V|log|V|)$. On the other hand, no such sorting is required in GCOMB. Thus, the speed-up achieved is higher in smaller budgets.

## 4.3 Performance on Influence Maximization

Influence Maximization (IM) is the hardest of the three combinatorial problems since estimating the spread of a node is #P-hard [14].

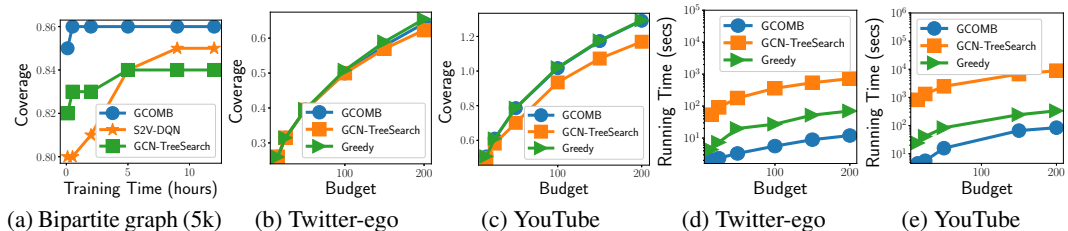

(a) Bipartite graph (5k)  (b) Twitter-ego  (c) YouTube  (d) Twitter-ego  (e) YouTube

Figure 2: MCP: (a) Improvement in Coverage against training time at $b = 15$. (b-c) Coverage achieved by GCOMB, GCN-TREESEARCH and Greedy. (d-e) Running times in TW-ew and YT.

**Edge weights:** We assign edge weights that denote the influence of a connection using the two popular models [2]: **(1) Constant (CO:)** All edge weights are set to 0.1, **(2) Tri-valency (TV):** Edge weights are sampled randomly from the set $\{0.1, 0.01, 0.001\}$. In addition, we also employ a third **(3) Learned (LND)** model, where we *learn* the influence probabilities from the action logs of users. This is only applicable to the Stack data which contain *action logs* from $8/2008$ to $3/2016$. We define the influence of $u$ on $v$ as the probability of $v$ interacting with $u$'s content at least once in a month.

**Train-Validation-Test split:** In all of the subsequent experiments, for CO and TV edge weight models, we train and validate on a subgraph sampled out of YT by randomly selecting $30\%$ of the edges ($50\%$ of this subset is used for training and $50\%$ is used for validation). For LND edge weight models, we train and validate on the subgraph induced by the $30\%$ of the earliest edges from Stack in terms of temporal order. While testing, on YT and Stack, we use the graph formed by the remaining $70\%$ of the edges that are not used for training. On other datasets, we use the entire graph for testing since neither those datasets nor their subsets are used for training purposes.

**GCOMB vs. GCN-TREESEARCH:** Fig. 3a compares the running time in IM on progressively larger subgraphs extracted from YT. While GCN-TREESEARCH consumes $\approx 3$ hours on the $70\%$ sub-graph, GCOMB finishes in 5 seconds.

**GCOMB vs. NOISEPRUNER+CELF** NOISEPRUNER+CELF, i.e., running CELF only on non-noisy nodes, is orders of magnitude slower than GCOMB in IM (See Fig 3d). Pruning noisy nodes does not reduce the graph size; it only reduces the number of candidate nodes. To compute expected spread in IM, we still require the entire graph, resulting in non-scalability.

**Billion-sized graphs:** IMM crashes on both the billion-sized datasets of TW and FS, as well as Orkut. Unsurprisingly, similar results have been reported in [2]. IMM strategically samples a subgraph of the entire graph based on the edge weights. On this sampled subgraph, it estimates the influence of a node using *reverse reachability sets*. On large graphs, the sample size exceeds the RAM capacity of 256GB. Hence, it crashes. In contrast, GCOMB finishes within minutes for smaller budgets ($b < 30$) and within 100 minutes on larger budgets of 100 and 200 (Figs. 3g-3h ). This massive scalability of GCOMB is a result of low storage overhead (only the graph and GCN and Q-learning parameters; detailed Space complexity provided in App. D in the Supplementary) and relying on just forwarded passes through GCN and $Q$-learning. The speed-up with respect to OPIM on billion-sized graphs can be seen in App. J.

**Performance on YT and Stack:** Since IMM crashes on Orkut, TW, and FS, we compare the quality of GCOMB with IMM on YT and Stack. Table 3a reports the results in terms of *spread difference*, where Spread Difference $= \frac{f(S_{IMM}) - f(S_{\text{Gcomb}})}{f(S_{IMM})} \times 100$. $S_{IMM}$ and $S_{\text{Gcomb}}$ are answer sets computed by IMM and GCOMB respectively. A negative spread difference indicates better performance by GCOMB. The expected spread of a given set of nodes $S$, i.e. $f(S)$, is computed by taking the average spread across $10,000$ Monte Carlo simulations.

Table 3a shows that the expected spread obtained by both techniques are extremely close. The true impact of GCOMB is realized when Table 3a is considered in conjunction with Figs. 3b-3c, which shows GCOMB is 30 to 160 times faster than IMM. In this plot, speed-up is measured as $\frac{time_{IMM}}{time_{\text{Gcomb}}}$ where $time_{IMM}$ and $time_{\text{Gcomb}}$ are the running times of IMM and GCOMB respectively.

Similar behavior is observed when compared against OPIM as seen in Table 3b and Figs. 3e- 3f.

| $b$ | YT-TV | YT-CO | Stack-TV | Stack-CO | Stack-LND | $b$ | YT-TV | YT-CO | Stack-TV | Stack-CO | Stack-LND |
|---|---|---|---|---|---|---|---|---|---|---|---|
| 10 | $-1 \times 10^{-3}$ | $1 \times 10^{-4}$ | $2 \times 10^{-5}$ | $\approx 0$ | $1 \times 10^{-5}$ | 10 | $-5 \times 10^{-5}$ | $-1 \times 10^{-5}$ | $2 \times 10^{-5}$ | $\approx 0$ | $1 \times 10^{-5}$ |
| 20 | $-2 \times 10^{-3}$ | $2 \times 10^{-4}$ | $3 \times 10^{-5}$ | $3 \times 10^{-5}$ | $-7 \times 10^{-5}$ | 20 | $-1 \times 10^{-4}$ | $1 \times 10^{-5}$ | $3 \times 10^{-5}$ | $2 \times 10^{-5}$ | $-2 \times 10^{-5}$ |
| 50 | $-3 \times 10^{-3}$ | $-5 \times 10^{-5}$ | $2 \times 10^{-5}$ | $6 \times 10^{-5}$ | $-7 \times 10^{-5}$ | 50 | $-2 \times 10^{-4}$ | $-3 \times 10^{-5}$ | $2 \times 10^{-5}$ | $5 \times 10^{-5}$ | $-6 \times 10^{-4}$ |
| 100 | $-1 \times 10^{-3}$ | $6 \times 10^{-4}$ | $2 \times 10^{-4}$ | $2 \times 10^{-4}$ | $-1 \times 10^{-4}$ | 100 | $-3 \times 10^{-4}$ | $2 \times 10^{-5}$ | $1 \times 10^{-4}$ | $7 \times 10^{-5}$ | $-2 \times 10^{-4}$ |
| 150 | $-6 \times 10^{-4}$ | $3 \times 10^{-4}$ | $1 \times 10^{-4}$ | $1 \times 10^{-4}$ | $-3 \times 10^{-5}$ | 150 | $-3 \times 10^{-4}$ | $-2 \times 10^{-5}$ | $1 \times 10^{-4}$ | $1 \times 10^{-4}$ | $-3 \times 10^{-4}$ |
| 200 | $-2 \times 10^{-3}$ | $2 \times 10^{-5}$ | $2 \times 10^{-4}$ | $2 \times 10^{-4}$ | $-1 \times 10^{-4}$ | 200 | $-4 \times 10^{-4}$ | $-7 \times 10^{-5}$ | $2 \times 10^{-4}$ | $2 \times 10^{-4}$ | $-3 \times 10^{-4}$ |

(a) Spread difference between IMM and GCOMB.     (b) Spread difference between OPIM and GCOMB.

Table 3: Comparison with respect to (a) IMM and (b) OPIM on YT and Stack. A *negative* value, highlighted in bold, indicates *better performance by* GCOMB.

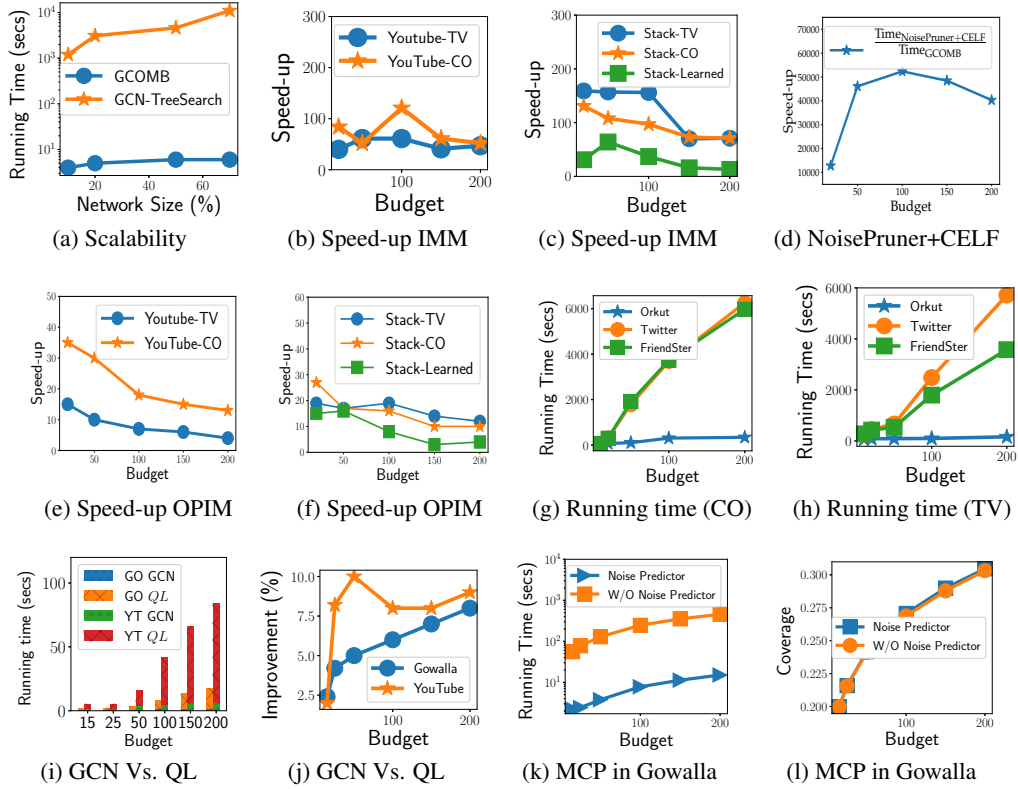

Figure 3: (a) Comparison of running time between GCOMB and GCN-TREESEARCH in YT at $b = 20$. (b-c) Speed-up achieved by GCOMB over IMM. (d) Speed-up achieved by GCOMB over NoisePruner+CELF on IM. (e-f) Speed-up achieved by GCOMB over OPIM. (g-h) Running times of GCOMB in IM in large graphs under the CO and TV edge models. (i) Distributions of running time between GCN and $Q$-learning in GO and YT datasets for MCP. (j) Improvement of $Q$-learning over GCN in MCP. (k-l) Impact of noise predictor on (k) running time and (l) quality.

## 4.4 Design Choices

**Impact of $Q$-learning:** Since GCN predicts the expected marginal gain of a node, why not simply select the top-$b$ nodes with the highest predicted marginal gains for the given budget $b$? This is a pertinent question since, as visible in Fig. 3i, majority of the time in GCOMB is spent on $Q$-learning. Fig. 3j shows that $Q$-learning imparts an additional coverage of up to $10\%$. Improvement ($\%$) is quantified as $\frac{Coverage_{\text{GCOMB}} - Coverage_{GCN}}{Coverage_{GCN}} \times 100$.

**Impact of Noise Predictor:** Fig. 3k presents the impact of noise predictor which is close to two orders of magnitude reduction in running time. This improvement, however, does not come at the cost of efficacy (Fig. 3l). In fact, the quality improves slightly due to the removal of noisy nodes.

## 5  Conclusion

S2V-DQN [7] initiated the promising direction of learning combinatorial algorithms on graphs. GCN-TREESEARCH [19] pursued the same line of work and enhanced scalability to larger graphs. However, the barrier to million and billion-sized graphs remained. GCOMB removes this barrier with a new lightweight architecture. In particular, GCOMB uses a phase-wise mixture of supervised and reinforcement learning. While the supervised component predicts individual node qualities and prunes those that are unlikely to be part of the solution set, the $Q$-learning architecture carefully analyzes the remaining high-quality nodes to identify those that collectively form a good solution set. This architecture allows GCOMB to generalize to unseen graphs of significantly larger sizes and convincingly outperform the state of the art in efficiency and efficacy. Nonetheless, there is scope for improvement. GCOMB is limited to set combinatorial problems on graphs. In future, we will explore a bigger class of combinatorial algorithms such as sequential and capacity constrained problems.

## Broader Impact

The need to solve NP-hard combinatorial problems on graphs routinely arise in several real-world problems. Examples include facility location problems on road networks [20], strategies to combat rumor propagation in online social networks [3], computational sustainability [8] and health-care [33]. Each of these problems plays an important role in our society. Consequently, designing effective and efficient solutions are important, and our current work is a step in that direction. The major impact of this paper is that good heuristics for NP-hard problems can be learned for large-scale data. While we are not the first to observe that heuristics for combinatorial algorithms can be learned, we are the first to make them scale to billion-size graphs, thereby bringing an algorithmic idea to practical use-cases.

## Acknowledgments and Disclosure of Funding

The project was partially supported by the National Science Foundation under award IIS-1817046. Further, Sahil Manchanda acknowledges the financial support from the Ministry of Human Resource Development (MHRD) of India and the Department of Computer Science and Engineering, IIT Delhi.

## Footnotes

\*denotes equal contribution

[2]We are, however, limited to set combinatorial problems only.

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
