[Supplementary Material]

# 6 Appendix

## A Influence Maximization

**Definition 1 (Social Network)** *A social network is denoted as an edge-weighted graph $G(V, E, W)$, where $V$ is the set of nodes (users), $E$ is the set of directed edges (relationships), and $W$ is the set of edge-weights corresponding to each edge in $E$.*

The objective in *influence maximization (IM)* is to maximize the *spread* of influence in a network through activation of an initial set of $b$ *seed* nodes.

**Definition 2 (Seed Node)** *A node $v \in V$ that acts as the source of information diffusion in the graph $G(V, E, W)$ is called a seed node. The set of seed nodes is denoted by $S$.*

**Definition 3 (Active Node)** *A node $v \in V$ is deemed active if either (1) It is a seed node ($v \in S$) or (2) It is influenced by a previously active node $u \in V_a$. Once activated, the node $v$ is added to the set of active nodes $V_a$.*

Initially, the set of active nodes $V_a$ is the seed nodes $S$. The spread of influence is guided by the *Independent Cascade (IC)* model.

**Definition 4 (Independent Cascade [14])** *Under the IC model, time unfolds in discrete steps. At any time-step $i$, each newly activated node $u \in V_a$ gets one independent attempt to activate each of its outgoing neighbors $v$ with a probability $p_{(u,v)} = W(u, v)$. The spreading process terminates when in two consecutive time steps the set of active nodes remain unchanged.*

**Definition 5 (Spread)** *The spread $\Gamma(S)$ of a set of seed nodes $S$ is defined as the total proportion of nodes that are active at the end of the information diffusion process. Mathematically, $\Gamma(S) = \frac{|V_a|}{|V|}$.*

Since the information diffusion is a stochastic process, the measure of interest is the *expected* value of spread. The *expected* value of spread $f(\cdot) = \mathbb{E}[\Gamma(\cdot)]$ is computed by simulating the spread function a large number of times. The goal in IM, is therefore to solve the following problem.

**Influence Maximization (IM) Problem [14]:** Given a budget $b$, a social network $G$, and a information diffusion model $\mathcal{M}$, select a set $S^*$ of $b$ nodes such that the expected diffusion spread $f(S^*) = \mathbb{E}[\Gamma(S^*)]$ is maximized.

## B The greedy approach

Greedy provides an $1 - \frac{1}{e}$-approximation for all three NP-hard problems of MCP, MVC, and IM[14]. Algorithm 1 presents the pseudocode. The input to the algorithm is a graph $G = (V, E)$, an optimization function $f(S)$ and the budget $b$. Starting from an empty solution set $S$, Algorithm 1 iteratively builds the solution by adding the "best" node to $S$ in each iteration (lines 3-5). The best node $v^* \in V \setminus S$ is the one that provides the highest *marginal gain* on the optimization function (line 4). The process ends after $b$ iterations.

**Limitations of greedy:** Greedy itself has scalability challenges depending on the nature of the problem. Specifically, in Alg. 1 there are two expensive computations. First, computing the optimization function $f(\cdot)$ itself may be expensive. For instance, computing the expected spread in IM is $\#P$-hard [2]. Second, even if $f(\cdot)$ is efficiently computable, computing the marginal gain is often expensive. To elaborate, in MCP, computing the marginal gain involves a setminus operation on the neighborhood lists of all nodes $v \notin S$ with the neighborhood of all nodes $u \in S$, where $S$ is the set of solution nodes till now. Each setminus operation consumes $O(d)$ time where $d$ is the average degree of nodes, resulting in a total complexity of $O(bd|V|)$. In IM, the cost is even higher with a complexity of $O(b|V|^2)$. In GCOMB, we overcome these scalability bottlenecks without compromising on the quality. GCOMB utilizes GCN [10] to solve the first bottleneck of predicting $f(\cdot)$. Next, a deep $Q$-learning network is designed to estimate marginal gains efficiently. With this unique combination, GCOMB can scale to billion-sized graphs.

---

**Algorithm 1** The greedy approach

---

**Require:** $G = (V, E)$, optimization function $f(.)$, budget $b$
**Ensure:** solution set $S$, $|S| = b$
  1: $S \leftarrow \emptyset$
  2: $i \leftarrow 0$
  3: **while** $(i < b)$ **do**
  4:    $v^* \leftarrow \arg\max_{\forall v \in V \setminus S} \{f(S \cup \{v\}) - f(S)\}$
  5:    $S \leftarrow S \cup \{v^*\}, i \leftarrow i + 1$
  6: **Return** $S$

---

---

**Algorithm 2** The probabilistic greedy approach

---

**Require:** $G = (V, E)$, optimization function $f(.)$, convergence threshold $\Delta$
**Ensure:** solution set $S$, $|S| = b$
  1: $S \leftarrow \emptyset$
  2: **while** $(gain > \Delta)$ **do**
  3:    $v \leftarrow$ Choose with probability $\frac{f(S \cup \{v\}) - f(S)}{\sum_{\forall v' \in V \setminus S} f(S \cup \{v'\}) - f(S)}$
  4:    $gain \leftarrow f(S \cup \{v\}) - f(S)$
  5:    $S \leftarrow S \cup \{v\}$
  6: **Return** $S$

---

## B.1 Training the GCN:

For each node $v$, and its $score(v)$, which is generated using probabilistic greedy algorithm, we learn embeddings to predict this score via a Graph Convolutional Network (GCN) [10]. The pseudocode for this component is provided in Alg. 3.

From the given set of training graphs $\{G_1, \cdots, G_t\}$, we sample a graph $G_i$ and a normalized budget $b$ from the range of budgets $(0, b_{max}^i]$, where $b_{max}^i = \max_{j=0}^m \left\{ \frac{|S_j^i|}{|V_i|} \right\}$. To recall, $S_j^i$ denotes the $j^{th}$ solution set constructed by probabilistic greedy on graph $G_i$. Further, quantity $r_{max}^b$ is computed from the set of training graphs and their probabilistic greedy solutions as described in § 3.2. It is used to determine the nodes which are non-noisy for the budget $b$.

For a sampled training graph $G_i$ and budget $b$, only those nodes that have a realistic chance of being in the solution set are used to train the GCN (line 2). Each iteration in the outer loop represents the *depth* (line 4). In the inner loop, we iterate over all nodes which are non-noisy and in their K-hop neighborhood (line 5). While iterating over node $v$, we fetch the current representations of $v$'s neighbors and *aggregate* them through a MEANPOOL layer (lines 6-7). Specifically, for dimension $i$, we have: $\mathbf{h}_N^k(v)_i = \frac{1}{|N(v)|} \sum_{\forall u \in N(v)} h_{u_i}^{k-1}$. The aggregated vector is next *concatenated* with the representation of $v$, which is then fed through a fully connected layer with *ReLU* activation function (line 8), where ReLU is the *rectified linear unit* $(ReLU(z) = max(0, z))$. The output of this layer becomes the input to the next iteration of the outer loop. Intuitively, in each iteration of the outer loop, nodes aggregate information from their local neighbors, and with more iterations, nodes incrementally receive information from neighbors of higher depth (i.e., distance).

At depth 0, the embedding of each node $v$ is $h_v^0 = \mathbf{x}_v$, while the final embedding is $\boldsymbol{\mu}_v = h_v^K$ (line 9). In hidden layers, Alg. 3 requires the parameter set $\mathbb{W} = \{\mathbb{W}^k, k = 1, 2, \cdots, K\}$ to compute the node representations (line 8). Intuitively, $\mathbb{W}^k$ is used to propagate information across different depths of the model. To train the parameter set $\mathbb{W}$ and obtain predictive representations, the final representations are passed through another fully connected layer to obtain their predicted value $score'(v)$ (line 10). Further, the inclusion of 1-hop neighbors($V^{g,1}$) of $V^g$ in line 9 and line 10 is only for the importance sampling procedure. The parameters $\Theta_G$ for the proposed framework are therefore the weight matrices $\mathbb{W}$ and the weight vector $\mathbf{w}$. We draw multiple samples of graphs and budget and minimize the next equation using Adam optimizer [15] to learn the GCN parameters, $\Theta_G$.

$$J(\Theta_G) = \sum_{\sim \langle G_i, b \rangle} \frac{1}{|V_i^g|} \sum_{\forall v \in V_i^g} (score(v) - score'(v))^2 \tag{4}$$

**Algorithm 3** Graph Convolutional Network (GCN)

---

**Require:** $G = (V, E)$, $\{score(v)$, input features $\mathbf{x}_v \; \forall v \in V\}$, budget $b$, noisy-node cut off $r^b_{max}$, depth $K$, weight matrices $\mathbb{W}^k$, $\forall k \in [1, k]$ and weight vector $\mathbf{w}$, dimension size $m_G$.
**Ensure:** Quality score $score'(v)$ for good nodes and nodes in their 1-hop neighbors
1: $\mathbf{h}_v^0 \leftarrow \mathbf{x}_v, \; \forall v \in V$
2: $V^g \leftarrow \{v \in V \mid rank(v, G) < r^b_{max}\}$
3: $V^{g,K} \leftarrow$ K-hop neighborhood of $V^g$
4: **for** $k \in [1, K]$ **do**
5:     **for** $v \in V^g \cup V^{g,K}$ **do**
6:         $N(v) \leftarrow \{u | (v, u) \in E\}$
7:         $\mathbf{h}_N^k(v) \leftarrow \text{MEANPOOL}\left(\{h_u^{k-1}, \forall u \in N(v)\}\right)$
8:         $\mathbf{h}_v^k \leftarrow ReLU\left(\mathbb{W}^k \cdot \text{CONCAT}\left(\mathbf{h}_N^k(v), h_v^{k-1}\right)\right)$
9: $\boldsymbol{\mu}_v \leftarrow \mathbf{h}_v^K, \; \forall v \in V^g \cup V^{g,1}$
10: $score'(v) \leftarrow \mathbf{w}^T \cdot \boldsymbol{\mu}_v, \; \forall v \in V^g \cup V^{g,1}$

---

In the above equation, $V_i^g$ denotes the set of good nodes for budget $b$ in graph $G_i$.

**Defining $\mathbf{x}_v$:** The initial feature vector $\mathbf{x}_v$ at depth 0 should have the raw features that are relevant with respect to the combinatorial problem being solved. For example, in Influence Maximization (IM), the summation of the outgoing edge weights of a node is an indicator of its own spread.

## C Q-learning

The pseudocode of the Q-learning component is provided in Algorithm 4.

**Exploration vs. Exploitation:** In the initial phases of the training procedure, the prediction may be inaccurate as the model has not yet received enough training data to learn the parameters. Thus, with $\epsilon = \max\{0.05, 0.9^t\}$ probability we select a random node from $C_t$. Otherwise, we trust the model and choose the predicted best node. Since $\epsilon$ decays exponentially with $t$, as more training samples are observed, the likelihood to trust the prediction goes up. This policy is commonly used in practice and inspired from bandit learning [7].

$n$-**step $Q$-learning:** $n$-step $Q$-learning incorporates delayed rewards, where the final reward of interest is received later in the future during an episode (lines 6-9 in Alg. 4). The key idea here is to wait for $n$ steps before the approximator's parameters are updated and therefore, more accurately estimate future rewards.

**Fitted $Q$-learning:** For efficient learning of the parameters, we perform *fitted Q-iteration* [28], which results in faster convergence using a neural network as a function approximator [25]. Specifically, instead of updating the $Q$-function sample-by-sample, fitted $Q$-iteration uses *experience replay* with a batch of samples. Note that the training process in Alg. 4 is independent of budget. The $Q$-learning component learns the best action to take under a given circumstance (state space).

## D Complexity Analysis of the Test Phase

For this analysis, we assume the following terminologies. $d$ denotes the average degree of a node. $m_G$ and $m_Q$ denote the embedding dimensions in the GCN and $Q$-learning neural network respectively. As already introduced earlier, $b$ denotes the budget and $V$ is the set of all nodes.

### D.1 Time Complexity

In the test phase, a forward pass through the GCN is performed. Although the GCN's loss function only minimizes the prediction with respect to the good nodes, due to message passing from neighbors, in a $K$-layered GCN , we need the $K$-hop neighbors of the good nodes (we will denote this set as $V^{g,K}$). Each node in $V^{g,K}$ draws messages from its neighbors on which first we perform MEANPOOL and then dot products are computed to embed in a $m_G$-dimensional space. Applying MEANPOOL consumes $O(dm_G)$ time since we need to make a linear pass over $d$ vectors of $m_G$ dimensions. Next, we perform $m_G$ dot-products on vectors of $m_G$ dimensions. Consequently, this consumes $O(m_G^2)$ time. Finally, this operation is repeated in each of the $K$ layers of the GCN. Since $K$ is typically 1 or 2, we ignore this factor. Thus, the total time complexity of a forward pass is $O(|V^{g,K}|(dm_G + m_G^2))$.

**Algorithm 4** Learning $Q$-function

---

**Require:** $\forall v \in V^g$, $score'(v)$, hyper-parameters $M$, $N$ relayed to fitted $Q$-learning, number of episodes $L$ and sample size $T$.
**Ensure:** Learn parameter set $\Theta_Q$
  1: Initialize experience replay memory $M$ to capacity $N$
  2: **for** episode $e \leftarrow 1$ to $L$ **do**
  3:    **for** step $t \leftarrow 1$ to $T$ **do**
  4:        $v_t \leftarrow \begin{cases} \text{random node } v \notin S_t \text{ with probability } \epsilon = \max\{0.05, 0.9^t\} \\ \text{argmax}_{v \notin S_t} Q'_n(S_t, v, \Theta_Q) \text{ otherwise} \end{cases}$
  5:        $S_{t+1} \leftarrow S_t \cup \{v_t\}$
  6:        **if** $t \geq n$ **then**
  7:            Add tuple $(S_{t-n}, v_{t-n}, \sum_{i=t-n}^{t} r(S_i, v_i), S_t)$ to $M$
  8:            Sample random batch $B$ from $M$
  9:            Update $\Theta_Q$ by Adam optimizer for $B$
 10: **return** $\Theta_Q$

---

The budget ($b$) number of forward passes are made in the $Q$-learning component over only $V^g$ (the set of good non-noisy nodes). In each pass, we compute locality and the predicted reward. To compute locality, we store the neighborhood as a hashmap, which consumes $O(d)$ time per node. Computing predicted reward involves dot products among vectors of $O(m_Q)$ dimensions. Thus, the total time complexity of the $Q$-learning component is $O(|V^g|b(d + m_Q))$.

For noise predictor, we need to identify the top-$l$ nodes based on $\mathbf{x_v}$ (typically the out-degree weight). $l$ is determined by the noise predictor as a function of $b$. This consumes $|V|log(l)$ time through the use of a min-Heap.

Combining all three components, the total time complexity of GCOMB is $O(|V|log(l) + |V^{g,K}|(dm_G + m_G^2) + |V^g|b(d + m_Q))$. Typically, $l << |V|$ (See Fig. 8a) and may be ignored. Thus, the total time complexity is $\approx O(|V| + |V^{g,K}|(dm_G + m_G^2) + |V^g|b(d + m_Q))$.

## D.2 Space Complexity

During testing, the entire graph is loaded in memory and is represented in linked list form which takes $O(|V| + |E|)$ space. The memory required for K layer GCN is $O(Km_G^2)$. Overall space complexity for GCN phase is $O(|V| + |E| + Km_G^2)$.

For the $Q$-learning component, entire graph is required for importance sampling purpose. It requires $O(|V| + |E|)$ space. Further, the space required for parameters for Q-network is $O(m_Q)$, since input dimension for Q-network is fixed to 2. Thus, space complexity of Q-network is $O(|V| + |E| + m_Q)$. Therefore, total space complexity of GCOMB is $O(|V| + |E| + Km_G^2 + m_Q)$.

## E  Number of parameters

**GCN:** If $m_G$ is the embedding dimension, each $\mathbb{W}_k$ is a matrix of dimension $m_G^2$. Other than $\mathbb{W}_k$, we learn another parameter $\boldsymbol{w}$ in the final layer (line 10 of Alg. 3) of $m_G$ dimension. Thus, the total parameter size is $K \times m_G^2 + m_G$, where $K$ is the number of layers in GCN.

**Q-learning:** If $m_Q$ is the dimension of the hidden layer in $Q$-learning, each of $\Theta_1$, $\Theta_2$, and $\Theta_3$ is a matrix of dimension $m_Q \times 2$. $\Theta_4$ is a vector of dimension $3m_Q$. Thus, the total number of parameters is $9m_Q$.

## F  Proof of Theorem 1

A sampling procedure is *unbiased* if it is possible to estimate the mean of the target population from the sampled population, i.e., $\mathbb{E}[\hat{\mu}_{N_z(V^g)}] = \mu_{N(V^g)} = \frac{\sum_{v \in N(V^g)} I(v)}{|N(V^g)|} = \frac{1}{|N(V^g)|}$ , where $\hat{\mu}_{N_z(V^g)}$ is the *weighted average* over the samples in $N_z(V^g)$. Specifically,

$$\hat{\mu}(N_z(V^g)) = \frac{1}{\sum_{v \in N_z(V^g)} \hat{w}_v} \sum_{v \in N_z(V^g)} \hat{w}_v \cdot I(v) \tag{5}$$

where $\hat{w}_v = \frac{1}{I(v)}$.

**Lemma 1** *Importance sampling is an unbiased estimate of $\mu_{N(V^g)}$, i.e., $\mathbb{E}\left[\hat{\mu}_{N_z(V^g)}\right] = \mu_{N(V^g)}$, if $\hat{w}_v = \frac{1}{I(v)}$.*

**Proof 1**

$$\mathbb{E}[\hat{\mu}_{N_z(V^g)}] = \frac{1}{\mathbb{E}[\sum_{v \in N_z(V^g)} \hat{w}_v]} \cdot \mathbb{E}\left[\sum_{v \in N_z(V^g)} \hat{w}_v \cdot I(v)\right]$$

*If we simplify the first term, we obtain*

$$\mathbb{E}\left[\sum_{v \in N_z(V^g)} \hat{w}_v\right] = z \times \mathbb{E}[\hat{w}_v]$$

$$= z \times \sum_{\forall v \in N(V^g)} \hat{w}_v \cdot I(v) = |N(V^g)| \times z$$

*From the second term, we get,*

$$\mathbb{E}\left[\sum_{v \in N_z(V^g)} \hat{w}_v \cdot I(v)\right] = z \times \mathbb{E}[\hat{w}_v \cdot I(v)] = z$$

*Combining these two, $\mathbb{E}[\hat{\mu}_{N_z(V^g)}] = \frac{z}{|N(V^g)| \times z} = \mu_{N(V^g)}$.*

Armed with an unbiased estimator, we show that a bounded number of samples provide an accurate estimation of the locality of a node.

**Lemma 2** *[**Theorem 1 in main draft**] Given $\epsilon$ as the error bound, $P\left[|\hat{\mu}_{N_z(V^g)} - \mu_{N(V^g)}| < \epsilon\right] > 1 - \frac{1}{|N(V^g)|^2}$, where $z$ is $O\left(\frac{\log |N(V^g)|}{\epsilon^2}\right)$.*

**Proof 2** *The samples can be viewed as random variables associated with the selection of a node. More specifically, the random variable, $X_i$, is the importance associated with the selection of the $i$-th node in the importance sample $N_z(V^g)$. Since the samples provide an unbiased estimate (Lemma 1) and are i.i.d., we can apply* Hoeffding's inequality *[11] to bound the error of the mean estimates:*

$$P\left[|\hat{\mu}_{N_z(V^g)} - \mu_{N(V^g)}| \geq \epsilon\right] \leq \delta$$

*where $\delta = 2\exp\left(-\frac{2z^2\epsilon^2}{\mathcal{T}}\right)$, $\mathcal{T} = \sum_{i=1}^{z}(b_i - a_i)^2$, and each $X_i$ is strictly bounded by the intervals $[a_i, b_i]$. Since we know that importance is bounded within $[0, 1]$, $[a_i, b_i] = [0, 1]$. Thus,*

$$\delta = 2\exp\left(-\frac{2z^2\epsilon^2}{z}\right) = 2\exp\left(-2z\epsilon^2\right)$$

*By setting the number of samples $z = \frac{\log(2|N(V^g)|^2)}{2\epsilon^2}$, we have,*

$$p\left[|\hat{\mu}_{N_z(V^g)} - \mu_{N(V^g)}| < \epsilon\right] > 1 - \frac{1}{|N(V^g)|^2}$$

## G   Training time distribution of different phases of GCOMB

Fig. 4 shows the distribution of time spent in different phases of training of GCOMB. Prob-Greedy refers to the phase in which probabilistic greedy algorithm is run on training graphs to obtain training labels for GCN component. Train-GCN and Train-QL refers to the training phases of GCN and Q-network respectively.

Figure 4: Phase-wise training time distribution of GCOMB

(a) Speed-up OPIM Orkut  (b) Speed-up OPIM Friendster

Figure 5: Speed up obtained by GCOMB over OPIM

## H Parameters

GCOMB has two components: GCN and the $Q$-Learning part. GCN is trained for $1000$ epochs with a learning rate of $0.001$, a dropout rate of $0.1$ and a convolution depth ($K$) of $2$. The embedding dimension is set to $60$. For training the $n$-step $Q$-Learning neural network, $n$ and discount factor $\gamma$ are set to $2$ and $0.8$ respectively, and a learning rate of $0.0005$ is used. The raw feature $x_v$ of node $v$ in the first layer of GCN is set to the summation of its outgoing edge weights. For undirected, unweighted graphs, this reduces to the degree. In each epoch of training, $8$ training examples are sampled uniformly from the Replay Memory with capacity $N = 50$ as described in Alg. 4. The sampling size $z$, in terms of percentage, is varied at $[1\%, 10\%, 30\%, 50\%, 75\%, 99\%]$ on the validation sets, and the best performing value is used. As we will show later in Fig.8c, $10\%$ is often enough.

For all train sets, we split into two equal halves, where the first half is used for training and the second half is used for validation. For the cases where we have only one training graph, like BrightKite(BK) in MCP and MVC, we randomly pick 50% of the edges for the training graph and the remaining 50% for the validation graph. The noise predictor interpolators in MCP are fitted on 10% randomly edge-sampled subgraphs from Gowallah, Twitter-ew and YouTube. During testing, the remaining 90% subgraph is used, which is edge disjoint to the 10% of the earlier sampled subgraph.

## I Extracting Subgraph from Gowalla

To extract the subgraph, we select a node proportional to its degree. Next, we initiate a breadth-first-search from this node, which expands iteratively till $X$ nodes are reached, where $X$ is the target size of the subgraph to be extracted. All of these $X$ nodes and any edge among these nodes become part of the subgraph.

## J Comparison with OPIM on billion sized graphs

Figs. 5a- 5b present the speed-up achieved by GCOMB over OPIM on Orkut and Friendster. Speed-up is measured as $\frac{time_{OPIM}}{time_{\text{GCOMB}}}$ where $time_{OPIM}$ and $time_{\text{GCOMB}}$ are the running times of OPIM and GCOMB respectively. OPIM crashes on Friendster-CO and Twitter dataset.

(a) Quality             (b) Scalability

Figure 6: MCP : Gowalla: a) Quality comparison of GCOMB and GCN-TREESEARCH against greedy. b) Running times of GCOMB and GCN-TREESEARCH against the greedy approach.

| Budget | Speed-up $\epsilon = 0.2$ | Coverage Difference $\epsilon = 0.2$ | Coverage Difference $\epsilon = 0.05$ |
|--------|---------------------------|--------------------------------------|---------------------------------------|
| 20 | 2 | −0.09 | −0.001 |
| 50 | 2 | −0.13 | −0.003 |
| 100 | 2 | −0.16 | −0.005 |
| 150 | 2 | −0.18 | −0.005 |
| 200 | 2 | −0.20 | −0.006 |

Table 4: Comparison with Stochastic Greedy(SG) algorithm. The $\epsilon$ parameter controls the accuracy of SG. A negative number means GCOMB is better than SG.

## K   Results on Max Cover Problem (MCP) on Gowalla

Fig. 6a presents the impact of budget on Coverage on Gowalla dataset. The quality achieved by GCOMB is similar to Greedy, while GCN-TREESEARCH is inferior. GCOMB is up to two orders of magnitude faster than GCN-TREESEARCH and 10 times faster than Greedy as can be seen in Fig. 6b.

## L   Comparison with Stochastic Greedy (SG) on MCP

We compare the performance of GCOMB with *SG* on MCP. As can be seen in Table 4, GCOMB is up to 20% better in quality at $\epsilon = 0.2$ and yet 2 times faster. SG fails to match quality even at $\epsilon = 0.05$, where it is even slower. Furthermore, SG is not drastically faster than CELF in MCP due to two reasons: (1) cost of computing marginal gain is $O(Avg.degree)$, which is fast. (2) The additional data structure maintenance in SG to access sampled nodes in sorted order does not substantially offset the savings in reduced marginal gain computations.

## M   Results on Max Vertex Cover (MVC)

To benchmark the performance in MVC, in addition to real datasets, we also use the Barabási–Albert (BA) graphs used in S2V-DQN [7].

**Barabási–Albert (BA):** In BA, the default edge density is set to 4, i.e., $|E| = 4|V|$. We use the notation BA-$X$ to denote the size of the generated graph, where $X$ is the number of nodes.

For synthetic datasets, all three techniques are trained on BA graphs with $1k$ nodes. For real datasets, the model is trained on Brightkite. Table 5 presents the coverage achieved at $b = 30$. Both GCOMB and GCN-TREESEARCH produce results that are very close to each other and slightly better than S2V-DQN. As in the case of MCP, S2V-DQN ran out of memory on graphs larger than BA-20k.

To analyze the efficiency, we next compare the prediction times of GCOMB with Greedy (Alg 1) and GCN-TREESEARCH. Figs. 7a-7c present the prediction times against budget. Similar to the results in MCP, GCOMB is one order of magnitude faster than Greedy and up to two orders of magnitude faster than GCN-TREESEARCH.

Figure 7: MVC: Running times of GCOMB and GCN-TREESEARCH against the greedy approach in (a) Gowalla (b) YouTube and c) Twitter-Ego.

| Graph | Greedy | S2V-DQN | GCN-TREESEARCH | GCOMB |
|---|---|---|---|---|
| **BA-10k** | **0.11** | 0.096 | 0.109 | **0.11** |
| **BA-20k** | **0.0781** | 0.0751 | **0.0781** | **0.0781** |
| **BA-50k** | **0.0491** | $NA$ | 0.0490 | **0.0491** |
| **BA-100k** | **0.0346** | $NA$ | 0.0328 | **0.0346** |
| **Gowalla** | **0.081** | $NA$ | **0.081** | **0.081** |
| **YouTube** | **0.060** | $NA$ | **0.060** | **0.060** |
| **Twitter-ego** | **0.031** | $NA$ | **0.031** | **0.031** |

Table 5: Coverage achieved in the Max Vertex Cover (MVC) problem. The best result in each dataset is highlighted in bold.

## N    Impact of Parameters

### N.1    Size of training data

In Fig. 8b, we evaluate the impact of training data size on expected spread in IM. The budget for this experiment is set to 20. We observe that even when we use only $5\%$ of YT to train, the result is almost identical to training with a $25\%$ subgraph. This indicates GCOMB is able to learn a generalized policy even with small amount of training data.

### N.2    Effect of sampling rate

We examine how the sampling rate in locality computation affects the overall coverage in MCP. In Fig. 8c, with the increase of samples, the accuracy at $b = 100$ increases slightly. At $b = 25$, the increase is negligible. This indicates that our sampling scheme does not compromise on quality.

### N.3    Dimension

We vary the GCN embedding dimension from $40$ to $160$ and measure its impact on coverage in MCP (Fig. 8d). We observe minute variations in quality, which indicates that GCOMB is robust and does not require heavy amount of parameter optimization.

(a) TW and FS     (b) Quality Vs. Training     (c) Gowalla     (d) Gowalla

Figure 8: (a) Number of nodes included in $V^g$ in Twitter and Friendster for different budgets $b$. (b) Impact of training set size on spread quality in IM. (c-d) Effect of sampling rate and embedding dimension across different budgets on MCP coverage.