[Reviews · NeurIPS 2020]

Review 1

Summary and Contributions: The paper introduces a new algorithm to learn combinatorial optimization algorithms on graphs. The method applies to the problems of maximum coverage and influence maximization. The new method is inspired and improves recent work on learning combinatorial optimization algorithms on graphs. It relies on techniques of graph embedding and reinforcement learning. The method is empirically compared with previous learning algorithms for combinatorial problems on graphs, as well as state-of-the-art combinatorial algorithms. The results show that the new method achieves solutions of same quality, but it is much faster.

Strengths: 1. The problems studied in the paper are interesting, and the techniques are relevant to the NeurIPS community. 2. The empirical results are quite convincing. In particular, there has been a lot of work on combinatorial algorithms for influence maximization, and improving over the state-of-the-art methods is a very good result.

Weaknesses: 1. The method is mainly heuristic, there is no guarantee for the performance of the new method. Accordingly the quality of the method can be judged only empirically on the datasets that have been tested. 2. I am not an expert in the are, but my impression is that the novelty of the work is somewhat limited. In particular the novelty is mainly to refine the framework of Dai et al. [4] on the particular problem, and to introduce the components of noise predictor and importance sampling for scalability.

Correctness: As far as I could check, the claims and the methods are correct.

Clarity: The paper is well written and easy to read.

Relation to Prior Work: To my knowledge, the related work is adequately discussed, and the main contributions are clearly presented.

Reproducibility: Yes

Additional Feedback: Post rebuttal: I thank the reviewers for their response.


Review 2

Summary and Contributions: The authors present a scalable learning-based heuristic approach for a set of hard cardinality-constrained combinatorial set problems on graphs, studying the empirical performance on cardinality constrained submodular maximization settings using real-world and synthetic data, and giving time and space complexities for the system’s various components. The approach seems to have better scalability than previous approaches for learning combinatorial algorithms on graphs. They achieve this scalability by having three sequential modules: - Pruning nodes in the input graph based on incident edge weight - A GCN that predicts a score related to an average of marginal gains from adding that vertex during collected probabilistic greedy solves - Q learning to iteratively add nodes to a solution that predicts the discounted value of adding that vertex given the current set of nodes in the solution as the state. Here the authors further improve runtime by computing a “locality” node feature based on sampling nodes according to their predicted scores from the GCN. The authors evaluate runtime and solution quality to compare their approach to the two learning-based approaches applicable for these problems, and a naïve greedy algorithm, demonstrating improved solution quality and faster runtimes compared to learning approaches and faster runtimes compared to the naïve greedy algorithm, sometimes with marginal solution quality improvement over greedy. The authors present experiments on maximum coverage on bipartite graphs, and influence maximization in the main text, and budget constrained vertex cover in the supplementary information. Additionally, the authors provide a small ablation study to understand the impact of using the Q learning module over just the GCN-based scoring module, as well as the impact on the pipeline from node pruning.

Strengths: The main strengths are in outperforming previous learning-based approaches, providing computational and space complexity, as well as providing a broad set of experiments in the specified domains with diverse settings of large real world and synthetic problems. Additionally, the authors provide a good motivation for their work and situate it well with respect to the relevant literature on learning for combinatorial optimization on graphs. Empirically, the approach seems to outperform previous learning-based approaches to solving combinatorial problems on graphs for cases where greedy algorithms yield a 1-1/e approximation. Additionally, even though there are several components to the proposed complex system, the different components are well motivated and the ablation study hints at all components being necessary for good performance. The method itself is well described and the authors provide relevant code and pseudocode in the appendix. The approach is relatively novel and adds supervised learning components, importance sampling, and node preprocessing to existing work in reinforcement learning for combinatorial optimization. Finally, the approach is relevant to NeurIPS as it approaches combinatorial optimization with a novel learning-based approach that incorporates domain-knowledge, and specialized methods to improve scalability and performance over existing learning-based approaches.

Weaknesses: The main weaknesses of the paper are that the work only uses a naïve version of the greedy algorithm rather than the faster lazy greedy algorithm, and that it seems to claim more than the results suggest without further investigation in terms of the scope of applicability, and performance improvements over the greedy algorithm. The approach seems to be specialized to selecting a set of elements for coverage-like problems and specifically submodular maximization problems which admit greedy approximation algorithms, not necessarily general set combinatorial problems as claimed (it is important to clearly and fairly articulate the claimed scope of the proposed algorithms superior performance). Additionally, the greedy algorithm empirically gives near-optimal performance in the experiments, so it would be useful to know whether this approach performs well for more difficult problems, where greedy is not almost optimal. It would be good to see performance on other more combinatorial problems or nonsubmodular set graph problems, e.g. picking a subset of nodes in a graph to allow spread for IC maximization instead of selecting seed nodes (Sheldon et al 2010) , which may not yield as easily to greedy algorithms. The score supervision used to train the GCN is highly related to the marginal return that greedy would use to score nodes. In addition, the locality metric seems to directly consider the percent of neighbors of a node which are not currently covered by a partial solution, which is directly related to the coverage problems considered in this work. The locality measure and marginal improvement scoring are both related to coverage-like problems but may be potentially less impactful for more combinatorial problems. All three domains are cardinality constrained, and not more generally budget-constrained problems with node weights, hence again it will be important to articulate that distinction or add experiments in weighted budgeted settings. It seems the main benefit for the overall goal of a high-quality fast heuristic is runtime improvement as performance improvements over greedy seem very marginal when they occur. However, the authors don’t compare against CELF (lazy greedy) which will have the same quality guarantees as greedy, but will have faster runtime as it will compute marginal gains for “noisy” nodes once then realistically never update them again. It remains to be seen whether the approach will perform well against this standard scalability method for cardinality constrained submodular maximization. CELF was introduced in Cost-effective Outbreak Detection in Networks, Leskovec et al KDD 2007. 3 domains, max vertex cover (MVC), influence maximization, and maximum coverage, are described but results are only given for influence maximization and maximum coverage in the main text with smaller solution quality improvement results on MVC reported in supplementary as GCOMB doesn’t improve as substantially over GCN-TreeSearch. It would be helpful to include all results in the main text to clearly state the performance improvement in the considered settings.

Correctness: The methods and complexity seem to be correct.

Clarity: Overall the work is clear, there are just a few minor comments I have to improve the clarity of the results and some of the notation. In the notation S is reused multiple times for different sets of nodes, for the most part this is ok and makes sense within the context. However, S with superscript and subscript is reused in probabilistic greedy and the GCN component with the indices having reversed meanings. Consider changing one of these to improve clarity. In description of training time cutoff, Line 245, it says all models are trained for 12 hours, but figure 4 in the appendix puts training time as taking less than 2 hours, it would help to clarify this. The plots and results are somewhat difficult to read. - First, given the small improvements numerically over baselines, with some relative improvements being bolded with values of 10^-5, it would help to give standard error metrics to determine how realistic it will be to expect these improvements. As the experiments are run 5 times, it should be possible to determine whether these improvements are due to noise or substantial improvement. - The bolding process in table 1 could be made more immediately clear by possibly bolding the three entries where the approach improves over greedy and adding an asterisk or other notation where GCOMB matches greedy. - It would be helpful to have runtime results in tables as well, to get a numerical sense of runtimes, given that it’s difficult to understand runtime for Orkut in the plots as the values are close to 0. This would also help with the small bars in 3g. - For figure 2 it would help to use consistent colors for GCN-TreeSearch as it is green in 2a and yellow everywhere else. It would also help to keep the same marker shapes and colors for the different approaches across plots where possible for quick comparison. - It is unclear why table 1a and 1b are in the same table as 1a describes dataset statistics, whereas 1b describes performance improvement.

Relation to Prior Work: The authors clearly state their relation to prior work in reinforcement learning for combinatorial optimization on graphs. They provide a novel approach that integrates supervised learning in the training procedure which is based on the domain knowledge that the greedy algorithm performs near-optimally on the instances in question. More broadly, for the problems investigated in this paper, it would be good to reference scalable approaches like CELF.

Reproducibility: Yes

Additional Feedback: Organization of results, it would be good to organize the results in a more cohesive manner that explicitly demonstrates the benefits in both runtime and solution quality. It might be possible to add timing results to the current performance tables. Given that this approach is used heavily for solving cardinality constrained submodular optimization problems over graphs, and the myopic greedy algorithm seems to perform best, it may be helpful to know how important the discount factor is in the reinforcement learning, compared to an approach that tries to just predict marginal gains given a vertex and the state of the graph. Additionally, it would be interesting to know the performance of using just the preliminary node pruning step in conjunction with a greedy algorithm, as reducing the size of the graph itself might improve performance, at the cost of the approximation guarantees. This will also help support the claim that the subsequent learning-based components are necessary. The MCP instances on real-world graphs seem to be the same as MVC instances but which are reduced to the MCP problem (generating a bipartite graph by copying the original graph and adding edges between the copies of originally connected nodes). Would it be possible to directly compare results on the original MVC problems in the MCP formulation? I am putting more discussion needed for the broader impact because negative ethical and societal implications are not discussed. Some ideas here are that the approach could be used for nefarious influence maximization such as spreading fake news or otherwise damaging information. However, the positive broader impacts are well-written. If the authors modify the following, I am happy to increase my score: - Explanation for why lazy greedy (CELF) would not be relevant to these performance results, or results of running CELF. - Evidence or explanation of why this approach would perform well for budget constrained set combinatorial problems other than cardinality constrained submodular maximization. Potentially modifying the described scope of where this approach is applicable to be for approaches where greedy performs near-optimally (or showing results on more domains that support the general claim). - Addressing minor points of clarity below. o Claim in lines 35-37 is saying they learn an approximation algorithm, the authors should consider changing to using a heuristic algorithm as there are no approximation guarantees. o Unclear why GCN is referenced for the pruning step, it seems there are no trainable network parameters in this step and the noise prediction seems to be based on a rule-based system based on summing node weights rather than learning a GCN. **** POST-REBUTTAL *** I appreciate the response from the authors and in particular providing empirical results that compare to SOTA baselines that convinced me to raise my evaluation score.


Review 3

Summary and Contributions: Contributions of this paper are summarized below: * Propose a new framework for combinatorial optimization (Gcomb), based on Graph Convolutional Network, aimed at scaling-up to billion-scale graph, and managing budget constraints. * Conduct experimental evaluation on standard combinatorial optimization problems (i.e., Vertex Cover, Maximum Coverage, and Influence Maximization) demonstrating that Gcomb outperforms S2V-DQN and Gcn-TreeSearch in terms of scalability and solution quality, and it is often competitive to and faster than baselines.

Strengths: <disclaimer: I am a novice in Deep Neural Networks, and so I was not able to judge of the details on existing DNN-style heuristics such as Gcn-TreeSearch and S2V-DQN.> * The motivation of this paper is clear: Scaling-up is a crucial issue given that existing approaches are limited to small-scale graphs, generalizability to various problems is important, and imposing (budget) constraints is crucial. * Superiority over Gcn-TreeSearch & S2V-DQN is quite convincing: Gcomb's scalability against massive-scale data and solution accuracy over these existing works were impressive and convincing for me, through three standard combinatorial optimization problems, i.e., Vertex Cover, Maximum Coverage, and Influence Maximization (though I was not able to judge of many experimental details regarding the comparison to S2V-DQN & Gcn-TreeSearch due to my lack of expertise in GCN).

Weaknesses: * I have several concerns that the experimental design is not convincing enough for demonstrating the superiority over baselines: The authors compare Gcomb to baseline algorithms experimentally to demonstrate Gcomb's scalability and accuracy. However, the implementation of the greedy algorithm used in this paper seems to be too naive, First, there are several generic techniques for scaling-up the greedy algorithm. One is LazyGreedy, which can detect and prune elements whose marginal gain is never significant, which does not affect the resulting solution quality; LazyGreedy can be >100 times faster than the naive greedy in practice (e.g., [Leskovec-Krause-Guestrin-Faloutsos-VanBriesen-Glance. KDD'07. Cost-effective Outbreak Detection in Networks]), and hence the claim that Gcomb is ~10 times faster than the greedy (e.g., in Lines 276, 586, and 598) is not convincing. Also, StochasticGreedy [Mirzasoleiman-Badanidiyuru-Karbasi-Vondrak-Krause. AAAI'15. Lazier Than Lazy Greedy] is proven to evaluate the object function at most O(n log ε^{-1}) times (for parameter some ε), which is much faster than LazyGreedy, with a slight decrease in objective value. I also have concerns for the choice/implementation of specific algorithms for each problem. ** Maximum Coverage: In Appendix B, the authors claim that greedy's time complexity is O(bd|V|), where b is budget, d is the average degree, and V is the ground set. However, it is well known that a slightly-modified greedy algorithm on Maximum Coverage runs in nearly-linear time (e.g., ~ O(d|V|) time); see, e.g., [Borgs-Brautbar-Chayes-Lucier. SODA'14. Maximizing Social Influence in Nearly Optimal Time] Simply using such algorithms (without LazyGreedy) would result in 100x speed-up for the case of b=100. ** Influence Maximization: IMM is a state-of-the-art algorithm of Influence Maximization (in 2015) in a sense that it samples the smallest number of RR samples with the *worst-case theoretical* guarantee on approximation accuracy. In practice, other existing memory-saving and time-efficient algorithms give reasonable-quality solutions similar to IMM; e.g., SKIM [Cohen-Delling-Pajor-Werneck. CIKM'14. Sketch-based Influence Maximization and Computation: Scaling up with Guarantees] can easily scale to billion-edge scale networks, and is a reasonable choice. Also, OPIM in [Tang-Tang-Xiao-Yuan. SIGMOD'18. Online Processing Algorithms for Influence Maximization], which is an improvement over IMM, has been shown to be up to 3 orders of magnitude faster than IMM. Therefore, the conclusion that Gcomb "improves upon the state-of-the-art algorithm for Influence Maximization" is not convincing.

Correctness: I have several concerns regarding the design of experimental comparison to baselines; please refer to the "Weakness" part.

Clarity: This paper is generally well-written and easy-to-follow, excepting minors issues as follows. * Bolding rule in tables is confusing: For many configurations in Table 1(b) and Table 2(a), the greedy achieves the best-quality solutions. Why not to use bold fonts for such results on greedy? Some missing citations: * Line 280 and 438: Neither ref. [11] or [2] does not give #P-hardness of computation of the influence spread, which is proved in the following article: [Chen-Wang-Wang. KDD'10. Scalable influence maximization for prevalent viral marketing in large-scale social networks] [Wang-Chen-Wang. DMKD'12. Scalable influence maximization for independent cascade model in large-scale social networks] * Line 429: [11] does not prove (1-1/e)-factor approximation for the greedy algorithm (but prove monotonicity and submodularity of the influence function), which is in the following article: [Nemhauser-Wolsey-Fisher. Mathematical Programming'78. An analysis of approximations for maximizing submodular set functions--I]

Relation to Prior Work: Discussion about existing works appears to be enough to justify the motivation of this paper (i.e., scaling-up, adaptivity to general combinatorial optimization problems, and budget constraints).

Reproducibility: Yes

Additional Feedback: --------AUTHOR FEEDBACK-------- I appreciate the authors' feedback. But I feel that it doesn't resolve my concerns due to the following reasons: * Comparison to (near)linear-time MCP algorithm is missing: The authors seem not to compare Gcomb to linear-time greedy (which is mentioned in my review) but to CELF & SG, which are still slow. (See also, e.g., Exercise 35.3-3 in Introduction to Algorithms) > R3C2 [Comparison to SODA 14]: My point is that the RIS algorithm in SODA'14 (and other RIS-based algorithms) solves an instance of Maximum Coverage in a greedy manner, and the proof of Theorem 3.1 (in SODA'14) shows that this can be done in linear time in the total size of data structures that RIS builds. * I expect the comparison to SKIM in Influence Maximization (as mentioned in my review). I keep the same score.


Review 4

Summary and Contributions: The paper develops a deep reinforcement learning algorithm for influence maximization and related coverage type problems. The method is compared with greedy and other state of the art methods, in terms of objective value and running time, and shows significant improvement

Strengths: The approach is pretty interesting. The paper also identifies several heuristics to improve the running time of individual steps in the training and reinforcement learning. The empirical results are pretty impressive

Weaknesses: Several parts of the paper are hard to follow, and there are some inconsistencies in the notions used

Correctness: Seems right, but some of the details are hard to verify

Clarity: Could be improved

Relation to Prior Work: There is a lot of work on scaling the greedy influence maximization algorithm, e.g., [Borgs et al., SODA 2014], [Cohen et al., KDD 2014], etc, which should be discussed, in addition to ref [22]. The authors should also consider the paper (M. Minutoliet al., "cuRipples: Influence maximizationon multi-GPU systems", ICS 2020), which improves on the IMM paper

Reproducibility: Yes

Additional Feedback: Intro: I am not sure why ref [2] is being cited when the influence maximization paper is first mentioned. This was first introduced in ref [11], and should be cited as well in that context Section 2: if the graphs are generated from a distribution D, does it make sense for the objective to be probabilistic? For instance, for the coverage problem, should the objective be to maximize the expected value of |X|/|B| (or some other suitable objective, which takes D into account)? Otherwise, there is a risk an algorithm might do ok on some instances, but not overall with respect to D. Also, the distribution D should be clarified. Is it specifying a distribution for the edges? Or nodes as well? If the distribution D is known, then this is not really an "unseen graph", as one could just optimize over the distribution, and there has been a lot of work on stochastic optimization. Section 3.2: the noise prediction is not very clear, since the node quality prediction is also doing the same thing, namely determining nodes which should be in the solution. A better motivation and discussion will be helpful Section 4.2: the choice of real graphs doesn't seem to be consistent with the initially described setup, in which there is a distribution D from which the graphs are picked. The performance results in Fig 2 are quite impressive Section 4.3: The results in Table 2(b) are pretty interesting ------------------------------------------------------------------ I have read the author response. They have addressed several of the review concerns, but the point about the distributional assumption is not very clear

[Author Response · NeurIPS 2020]

We are addressing only the major comments in this document. Most certainly, we will incorporate all minor comments
relating to presentations, citations, clarifications, etc. In this document, RXCY refers to Comment Y by Reviewer X.
**R1C1 [Heuristic method]:** We agree GCOMB is a heuristic. This holds true for [4],[15] as well and, in general, for
most deep-learning techniques. Our main objective is to scale to billion-sized graphs, where [4],[15], and approximation
algorithms, such as Greedy, IMM and CELF fail. We will ensure to make this crystal clear.
**R1C2 [Novelty]:** GCOMB focuses on the *budget-constrained* scenario and this leads to several design choices that are
different from [4]. Examples include a novel *probabilistic greedy* algorithm to construct budget-focused training data,
and noise predictor for pruning the search space. In addition GCOMB uses a mixture of supervised learning (GCN) and
reinforcement learning. In contrast, [4] is an end-to-end reinforcement learning architecture and thus time-consuming.
**R2C1 [Relevance of CELF]:** As reported in Table 1, GCOMB is consistently faster than CELF in both MCP and IM.
Since CELF did not finish in IM even after 2 days, the reported results for IM are based on a random sample containing
5% of the edges in YouTube. The slowness of CELF in IM is also reported in [2].
**R2C2 [Noise pruning + GREEDY/CELF]:** NOISEPRUNER+CELF, i.e., running CELF only on non-noisy nodes, is
orders of magnitude slower than GCOMB in IM (See Table 1). Pruning noisy nodes does not reduce the graph size; it
only reduces the number of candidate nodes. To compute expected spread in IM, or coverage in MCP, we still require
the entire graph, resulting in non-scalability. For ex., at budget $b \geq 100$, CELF did not finish in 2 days as computing
marginal gain is #P-hard. Note that NOISEPRUNER+CELF is much faster than CELF. However, speed-up over CELF in
Table 1 is smaller since the dataset there is 5% of YouTube. If our accepted, we will use a common dataset for both
CELF and NOISEPRUNER+CELF, so that the speed-up is better depicted (We could not due to 5-day rebuttal deadline).
**R2C3 [Budget constrained set combinatorial problems]:** This is an excellent point. We propose to restate our
scope as follows: *We aim to learn algorithms that can be approximated well by greedy algorithm (or where greedy*
*is nearly optimal) and that have cardinality constraints.* Although GCOMB may generalize to other combinatorial
problems, it remains to be investigated. Learning with a small budget and generalizing it to a larger one in test phase is
quite non-trivial and we tackle it via probabilistic greedy and noise predictor.
**R2C4 [Approx. guarantees]:** See R1C1. **R2C5 [MVC in MCP formulation]** Will do. We lack space in rebuttal.
**R2C6 [Is NOISEPRUNER part of GCN]:** The noise prediction is a pre-processing step for GCN. As highlighted in
Fig. 1, the hidden layers of GCN are trained only with nodes that pass the noise filter. The noise prediction component
has $r^b_{max}$ as a trainable parameter, which is trained using the first layer information of GCN through linear interpolation.
**R3C1 [Compare with LazyGreedy (CELF), STOCHASTICGREEDY(SG), and OPIM]:** GCOMB is faster than all
three techniques (Table 1). For CELF, see R2C1 and R2C2. GCOMB is faster than OPIM and SG, and on average,
better in quality. Like CELF, SG did not finish in IM even after 10 hours as it only reduces the number of marginal cost
computations, but not the cost of computing the marginal gains itself, which is #P-hard. In contrast, GCOMB reduces
*both* number of marginal cost computations (Noise pruning) and cost of computing marginal gains (GCN+Q-learning).
In MCP, GCOMB is up to 20% better in quality at $\epsilon = 0.2$ and yet 2 times faster. SG fails to match quality even at
$\epsilon = 0.05$, where it is even slower. Furthermore, SG is not drastically faster than CELF in MCP due to two reasons: (1)
cost of computing marginal gain is $O(Avg.degree)$, which is fast. The additional data structure maintenance in SG to
access sampled nodes in sorted order does not substantially offset the savings in reduced marginal gain computations.
(2) Due to being scale-free networks, the inner loop of CELF terminates early in the sorted order.
**R3C2 [Comparison to SODA 14]:** IMM is an optimized version of the SODA-14 paper. SODA-14 does not scale due
to the $\epsilon^{-3}$ factor and other large hidden constants in its time complexity of $O(bl^2(|E| + |V|) \log^2 |V|/\epsilon^3)$.
**R4C1 [Citations]:** We will replace/add the citations as suggested.
**R4C2 [Explanation of distribution D] and R4C4 [Choice of real graphs]:** We do not assume any known distribution.
We only assume that the training and test graphs come from the same (unknown) distribution D. For example, if one
trains on scale-free social networks, the test graphs are expected to be scale-free social networks as well. We hope this
also clarifies our choice of real graph datasets, all of which are social networks. The metric $|X|/|B|$ is distribution
agnostic. We have reported the mean. We can also report the standard deviation through error bars.
**R4C3 [Noise prediction and node quality prediction]:** The key differentiating factor between noise predictor and
node quality prediction is their computation complexities. This design choice is motivated by the observation that
computationally expensive predictive tasks should be performed only for the promising nodes. Thus, the noise predictor
prunes out nodes that will *not* be in the solution. For the remaining nodes, node quality prediction is performed.

| Experiment | Speedup over CELF | | NOISEPRUNER+CELF | Speedup over OPIM | | Spread Difference with OPIM | | SG (MCP in YT) | | |
|---|---|---|---|---|---|---|---|---|---|---|
| Budget | IM (CO) in YT 5% | MCP in YT | Speedup in IM (CO) in YT | CO in YT | TV in YT | CO in YT | TV in YT | Speedup $\epsilon = 0.2$ | Coverage Difference $\epsilon = 0.2$ | Coverage Difference $\epsilon = 0.05$ |
| 20 | 9 | 4 | 12840 | 35 | 15 | $1 \times 10^{-5}$ | $-1 \times 10^{-4}$ | 2 | $-0.09$ | $-0.001$ |
| 50 | 11 | 4 | 46119 | 30 | 10 | $-3 \times 10^{-5}$ | $-2 \times 10^{-4}$ | 2 | $-0.13$ | $-0.003$ |
| 100 | 28 | 3 | – | 18 | 7 | $2 \times 10^{-5}$ | $-3 \times 10^{-4}$ | 2 | $-0.16$ | $-0.005$ |
| 150 | – | 2 | – | 15 | 6 | $-2 \times 10^{-5}$ | $-3 \times 10^{-4}$ | 2 | $-0.18$ | $-0.005$ |
| 200 | – | 2 | – | 13 | 4 | $-7 \times 10^{-5}$ | $-4 \times 10^{-4}$ | 2 | $-0.20$ | $-0.006$ |

Table 1: **[R2C1, R2C2, R3C1]** OPIM (version OPIM-$C^+$, which is the fastest. Code provided by authors) is run with the default parameters recommended by the authors in the paper. $\epsilon$ is recommended to be kept in range $[0.01, 0.1]$. Thus, we set it to $\epsilon = 0.05$. In Spread Difference and Coverage Difference, we compute (Baseline Performance-GCOMB performance), and thus, a negative difference indicates *better* performance by GCOMB. CO and TV denote the edge weight models in IM. YT denotes the YouTube dataset. MCP denotes Max Coverage Problem and IM denotes Influence Maximization.

[Meta-Review · NeurIPS 2020]

Three reviewers rated this paper as weak accept, and one as reject. All reviewers felt the paper combined learning-based techniques effectively to achieve impressive performance on combinatorial optimization problems in massive graphs. Reviewers describe the work as a combination of heuristics and modules consisting of existing techniques, but largely view the overall system as being significant, and comment on its impressive performance and an ablation study to justify individual components. The main criticisms were about missing comparisons to baselines. It was observed that the proposed method essentially does well on submodular coverage style problems where the greedy algorithm is often nearly optimal in practice and its main advantage is being much faster. The reviewers pointed out that a number of faster variants of the greedy algorithm are available and requested comparisons to these. In the rebuttal, the authors show significant gains relative to three additional baselines: CELF, SG, and OPIM. This was enough to convince R2 to raise their score. R3 felt that at least one relevant baseline was still missing. Some ambiguity remains here, but overall the meta-reviewer feels the authors provided significant evidence that their approach performs very well relatively to a range of baselines on a number of different problems. The authors are encouraged to improve the paper with the contents of the rebuttal and also consider the remaining comments of R3.